# Beyond the In-Practice CBC: The Research CBC Parameters-Driven Machine Learning Predictive Modeling for Early Differentiation among Leukemias

**DOI:** 10.3390/diagnostics12010138

**Published:** 2022-01-07

**Authors:** Rana Zeeshan Haider, Ikram Uddin Ujjan, Najeed Ahmed Khan, Eloisa Urrechaga, Tahir Sultan Shamsi

**Affiliations:** 1Baqai Institute of Hematology, Baqai Medical University, Karachi 75340, Pakistan; 2National Institute of Blood Disease (NIBD), Karachi 75300, Pakistan; 3Department of Pathology, Liaquat University of Medical and Health Sciences, Jamshoro 76090, Pakistan; ikramujjan1973@yahoo.es; 4Department of Computer Science, NED University of Engineering and Technology, Karachi 75270, Pakistan; najeed@neduet.edu.pk; 5Core Laboratory, Galdakao-Usansolo Hospital, 48960 Galdakao, Spain; eloisamaria.urrechagaigartua@osakidetza.eus

**Keywords:** leukemia, complete blood cell count, cell population data, CBC research parameters, machine learning, artificial neural network

## Abstract

A targeted and timely treatment can be a beneficial tool for patients with hematological emergencies (particularly acute leukemias). The key challenges in the early diagnosis of leukemias and related hematological disorders are their symptom-sharing nature and prolonged turnaround time as well as the expertise needed in reporting confirmatory tests. The present study made use of the potential morphological and immature fraction-related parameters (research items or cell population data) generated during complete blood cell count (CBC), through artificial intelligence (AI)/machine learning (ML) predictive modeling for early (at the pre-microscopic level) differentiation of various types of leukemias: acute from chronic as well as myeloid from lymphoid. The routine CBC parameters along with research CBC items from a hematology analyzer in the diagnosis of 1577 study subjects with hematological neoplasms were collected. The statistical and data visualization tools, including heat-map and principal component analysis (PCA,) helped in the evaluation of the predictive capacity of research CBC items. Next, research CBC parameter-driven artificial neural network (ANN) predictive modeling was developed to use the hidden trend (disease’s signature) by increasing the auguring accuracy of these potential morphometric parameters in differentiation of leukemias. The classical statistics for routine and research CBC parameters showed that as a whole, all study items are significantly deviated among various types of leukemias (study groups). The CPD parameter-driven heat-map gave clustering (separation) of myeloid from lymphoid leukemias, followed by the segregation (nodding) of the acute from the chronic class of that particular lineage. Furthermore, acute promyelocytic leukemia (APML) was also well individuated from other types of acute myeloid leukemia (AML). The PCA plot guided by research CBC items at notable variance vindicated the aforementioned findings of the CPD-driven heat-map. Through training of *ANN* predictive modeling, the CPD parameters successfully differentiate the chronic myeloid leukemia (CML), AML, APML, acute lymphoid leukemia (ALL), chronic lymphoid leukemia (CLL), and other related hematological neoplasms with AUC values of 0.937, 0.905, 0.805, 0.829, 0.870, and 0.789, respectively, at an agreeably significant (10.6%) false prediction rate. Overall practical results of using our *ANN* model were found quite satisfactory with values of 83.1% and 89.4.7% for training and testing datasets, respectively. We proposed that research CBC parameters could potentially be used for early differentiation of leukemias in the hematology–oncology unit. The CPD-driven ANN modeling is a novel practice that substantially strengthens the predictive potential of CPD items, allowing the clinicians to be confident about the typical trend of the “disease fingerprint” shown by these automated potential morphometric items.

## 1. Introduction

Artificial intelligence (AI) has seen remarkable development and increased value during the past two decades along with its successful introduction for solving complex data-related problems [1]. In fact, in recent years, a significant increase in this trend has been noted for the use of AI in clinical aims for all three conventional medical tasks: diagnosis, therapy, and prognosis, but comparatively more for diagnosis [2,3]. A well-known and commonly used process or tool thorough which *AI* can achieve its learning objectives is Machine Learning (ML). For clinicians, a major question is whether ML will prove itself as an applied clinical tool in medical diagnostics. Through a brief literature survey, we can easily find various studies regarding successful applications of the ML approach in specialized diagnostic fields [4,5,6,7,8]. However, for complex fields of clinical diagnostics such as hematology, limited examples of successful applications of ML have been reported.

For medical diagnoses where a physician needs to determine which disease most likely explains the patient’s signs and symptoms, personal medical skills, knowledge, and experience are interlinked and together play a key role [9]. Laboratory tests assist in the confirmation, exclusion, classification, and/or monitoring of diseases along with further guidance for treatment [10]. Sadly, the typical practice of clinicians and diagnostic staff is aimed exclusively at targeting ‘out of reference range’ parameters (values) in the entire test report, which can limit the extended and even actual diagnostic potential of laboratory tests. This approach has a direct link with the underestimation and overlooking of the actual power of diagnostic test results [11]. Heavy workloads and a lack of convenient solutions for assistance in the prompt screening of the entire details generated along with routine laboratory testing may be behind this current clinical/diagnostic practice. The diagnostic workup for hematological diseases, especially neoplasms like leukemias, are ordinarily led by a basic blood test: complete blood cell count (CBC), and even a well-trained technologist/hematologist can pass over the trends, relations, and deviations among the increasing values of classical and additional (extended) data/parameters generated by modern hematology analyzers. The particularly advanced hematologic analyzers are now able to count and recognize the morphological characteristics of the types/subtypes of blood cell subpopulations by using the principles of electrical impedance, radiofrequency conductivity, light scattering, and/or Cytochemistry [12]. The above-mentioned analyzers also incorporate innovative computer algorithms and advanced hardware technology that help them in the collection and generation of the cell’s morphologic data, which are called Research CBC parameters or cell population data (CPD), a potential automated quantitative morphologic item. The Research CBC parameters are generated by multiple channels corresponding to the size, complexity (cytoplasmic granularity), and DNA/RNA content [12]. The presence of immature/abnormal white blood cells (WBCs) in peripheral blood deviate the values of CPD parameters, thus able to offer some sort of ‘disease signature’. The degree of modification in the values of CPD parameters is linked with the number and immaturity or abnormality of that individual type of immature/abnormal white blood cell (WBCs). The clinical utility of these research CBC parameters, particularly for sepsis, infection, and hematological disorders, are well reported [13,14,15,16,17,18,19,20,21,22,23,24,25,26,27]. Morphological assessments through these CPD items might be preeminently objective, quantitative, and automated and can minimize the risk of subjective interpretation [22,28].

Compared to other hematological disorders, leukemias have highly bizarre clinical, morphological, and biological characteristics. Starting treatment without waiting for a definitive diagnosis or delaying for other concerns is an effective practice in treating leukemias [29,30]. In routine practice in a hematology clinic, ML tools can effectively facilitate the clinical personals by smartly handling hundreds of attributes (items) like CPD parameters, and they have the potential for sensible detection and utilization of various patterns (disease’s fingerprints) among routine and research CBC parameters for various diseases. The particular disease signatures can make this specialization of diagnostics especially tempting for AI/ML applications. Apart from the morphology-based classification systems and their limitations, we worked to propose a novel prediction model based on the ML framework that would utilize the deviation trends among the values of CBC data, especially CPD parameters, as ‘fingerprints’ for leukemias and related disorders. This CBC data-driven ML (artificial neural network (ANN)) tool will offer efficient screening for backing early expulsion and directing presentation of “patients’ flow-from” in hematology–oncology departments.

## 2. Material and Methods

### 2.1. Study Population

The study population consisted of newly diagnosed acute leukemia cases that were presented to the academic research center (National Institute of Blood Diseases and Bone Marrow Transplantation, Karachi, Pakistan) from February 2014 to December 2020. A total of 1577 cases having >1000 WBCs and at least a 2% immature/abnormal WBCs cell count in peripheral blood with a complete diagnostic work-up are included in this study. The diagnosis and subtypes of leukemias and related hematological disorders were confirmed through bone marrow examination, immunophenotyping, cytogenetic and/or molecular tests based on initial workups, along with clinical and demographic information according to the WHO classification of tumors of the hematopoietic and lymphoid tissues (2008). In the final reports from diagnostic work-ups, all cases were allocated to one of the six principle disease groups and thus 354, 96, 213, 272, 153, and 489 cases were of acute myeloid leukemia (AML), acute promyelocytic leukemia (APML) PML-RARA, chronic myeloid leukemia (CML), acute lymphoid leukemia (ALL), chronic lymphoid leukemia (CLL), and ‘others (’Non-Hodgkin’s lymphoma, Plasma cell dyscrasia, and etc.), respectively. The collection of patient data and samples (blood and bone marrow) has been carried out in accordance with the Declaration of Helsinki, under the terms of all relevant local legislation. The responsible ethical committee of the National Institute of Blood Disease (NIBD) reviewed and approved the study in accordance with the ‘medical research involving human subjects act’ on permit number: NIBD/RD-167/14-201 dated 16th December 2013. Each study subject gave informed consent.

### 2.2. Sample Preparation and Methods

Overall, 1577 peripheral whole blood samples were collected in K3EDTA blood tubes (Becton Dickinson, Franklin Lakes, NJ, USA). The analysis of all samples was performed with the Sysmex XN-Module (Kobe, Japan) by strictly following the manufacturer’s instructions, and the quality of the data was validated by regular analysis of internal quality control material (XN-CHECK levels 1, 2 and 3; Streck Laboratories Inc., Omaha, NE, USA). A peripheral blood smear was also prepared for all samples followed by May–Grunwald-–Giemsa staining. Peripheral blood morphological examination was performed by optical microscopy (OM), in accordance with the recommendations of the International Council for Standardization in Hematology (ICSH) [31]. Briefly, differential count analysis by OM was carried out by two skilled hematologists (the opinion of a third hematologist was called upon where results were found to have >5% disagreement) on 200 cells at 100× magnification, as recommended by the CLSI document H20-A2 [32] and by the ICSH guidelines [31].

### 2.3. Classical Statistical Data Analysis

Data was analyzed using SPSS version 23.0 (New York, NY, USA) and visualized through Clustvis (Institute of Computer Science, University of Tartu, J. Liivi-Tartu, Estonia), which is a web tool for visualizing the clustering of multivariate data (inspired by the PREDECT project and mostly based on BoxPlotR codes). The calculation of mean, standard deviation (SD), and significance (*P*-) values among study groups were also carried out through SPSS.

To delve into and obtain visualization of the subtle patterns of the Research CBC parameters among study groups, heat map (a supervised data visualization tool) and principal component analyses (PCA) were conducted. Heat map and PCA plots were generated through “Clustvis” https://biit.cs.ut.ee/clustvis/ (accessed on 1 December 2021), which is a web tool for visualizing the clustering of multivariate data. Aimed clustering (nodding) of study parameters, the function ‘correlation’ for the “clustering distance”, ‘average’ for the “clustering methods” and ‘tightest cluster first’ considering the “tree ordering of columns” were used. For the color grading scheme of the heat map, the command of function ‘diverging: RdBu (Red to Blue)’ at ‘minimum-2 to maximum-2′ was used. The diverging: RdBu contains diverging palette options, more suitable for data with both negative and positive values, which is the same as in our data.

### 2.4. AI Based Approach

For AI applications, artificial neural network (ANN) from ML tools was selected for predictive modeling. The primary reason for selecting ANN is its superiority over other AI tools for finding patterns that are far too complex or numerous for a human programmer to extract and teach the machine to recognize. To conduct ANN predictive modeling with the aim of prediction/differentiation and classification, and to determine the nature of our dataset, first we tried the two most fitted modeling tools: Radial basis function network (RBFN) and Multiple perceptron (MLP) networking. In the training and testing stages, RBFN out-performed MLP with ‘lower percent incorrect prediction’ rates, so we continued with RBFN. It is a computational non-linear data modeling having three (input, hidden, and output) layers and a feed-forward, supervised learning network that can smartly classify the cases through the input layer (variables—in our case, CBC parameters) similarity measurement with respect to examples from a training (data) set. Each hidden layer stores a ‘prototype’ that is an individual example of many more present in the training set. For the classification of a new case, each variable computes the Euclidean distance among a new input and its prototype. An input layer/factor (which provides information from the outside world to the network), a hidden layer also called the radial basis function layer (which has no direct connection with the outside world and performs computations and transfers information from input nodes to output nodes), and an output layer/dependent variables (which is responsible for computations and transferring information from the network to the outside world). The hidden layer transforms the input vectors into radial basis functions.

**RBFN algorithm.** To create an RBFN predictive model, SPSS syntax-programming language was used that allowed operators for any possible modification, and in this way, we tried various dataset partitions of 50, 60, 70 and 50, 40, 30 for training and testing, respectively, for our network. In this regard, cases were randomly assigned based on relative numbers of cases without using any portioning variable to assign cases. In the architecture of our predictive network, we used options to automatically compute ranges in finding the best number of units within this range, and for numbers of units in the hidden layer, the normalized radial basis function was selected for the activation function of the hidden layer, and we tabbed an automatic computation of the amount of overlap to allow for overlapping among hidden units. Methodologically, the RBFN model was developed in two steps: first, by using clustering methods, radial basis functions were determined and the width and center of each of the radial basis functions were calculated. In the second step, the network determined the synaptic weights given the radial basis functions. Both classification and prediction through the output layer were laid by sum-of-squares error functions with the identity activation function.

We estimated the importance of set of the selected CBC attributes with our normalized radial basis function algorithm and evaluated our ML network results for classification (diagnostic) problems by observing the classification accuracy (true positive rate) generated after training the network against relevant data set. To avoid false negative and false positive results that may put the patient’s life at risk by causing delays in treatment as well as potentially resulting in unnecessary health care costs, we added a normal (healthy) population in the training set of our data. Furthermore, to visualize the network behavior, an ROC curve, predicted-by-observed chart, cumulative gains chart, and lift chart were generated.

## 3. Results

Baseline characteristics (routine CBC along with research items) of the analytic cohort in consonance with study groups are presented in Table 1 as mean, standard deviation (SD), and significance. Exceptionally, NRBC (%) was noted as insignificant while all other study parameters showed significant difference by classical statistical analysis. Furthermore, correlation-based clustering of study groups on the heat map (Figure 1) underpinned the subtle trends of CBC research items for the types of leukemias.

In the heat map illustration (Figure 1), color-grading the parameters (rows) allows a quick view of hot and cold spots within the dataset but also clusters (rearranges) the study groups (columns) with identical patterns by nodding (branching) them. As a whole, hot spots were noted for CML followed by AML and APML while CLL followed by ALL presented greater numbers of cold spots. Notably, group ‘others’ showed a mixed pattern (both cold and hot spots).

The nodding trends help us to find how closely patterned to each other our study parameters are. The step/level of any particular node where it groups to other node/s describes its degree of clustering (correlation, in our case). The first step of nodding was observed between CML and all other study groups. At the second step (level of nodding), AML and APML were separated from ALL, CLL, and group ‘others’. Through the third level, nodding group ‘others’ were distinct from ALL and CLL. Various levels of clustering at our Research CBC parameter driven heat map suggested the predictive potential of our study parameters, particularly for CML from other leukemias, as well as for differentiation of myeloid from lymphoid leukemias. The lower the level of nodding, the closer the values of the study (lower predictive potential).

The Principal Component Analysis (PCA) plot (Figure 2) was used to graphically visualize our multivariate data and to extract the key information as a set of new attributes called principle components (PC1 and PC2). PC1 and PC2 correspond to a linear merger of our Research CBC parameters aimed at the identification of directions/principal components along which the variation in the data is maximal. For the most part, the PCA plot vindicates the impression from heat map that CML from other study groups is highly distinct while myeloid vs. lymphoid leukemias are decidedly apparent from each other. Barely a minor degree of overlaps was noted for AML with APML, and ALL with group ‘others’ and CLL.

The model summary of our CBC research-driven RBFN predictive framework in Figure 3 presents various positive signs. The sum of squared error remained 46.59 and 22.18 for training and testing, respectively. The smaller value of the squared error in testing over training indicates the most-fit number of hidden units/layers to minimize error functions. Furthermore, the percentage of incorrect predictions was noticeably lower at 16.9 for training and 10.6 in consideration of the testing dataset. Additionally, practical results of using the network as shown in the classification table likewise decidedly remained denoting. The model’s performance-related scales in terms of predictive-pseudo probability, sensitivity and specificity (AUC), gain, and lift charts were found to be promisingly convincing for the predictive ability of the network. The ROC curve gives more powerful and much cleaner visual presentation of the specificity and sensitivity in a single plot than the series of tables. The ROC chart presents all six curves of AML, APML, CML, ALL, CLL, and group ‘others’ with area values of 0.810, 0.789, 0.937, 0.829, 0.905, and 0.805, respectively. In the predicted-by-observed chart, the ‘observed response’ and ‘predicted categories’ were aligned with the x-axis and y-axis, respectively. The prediction is considered ‘Correct’ when the boxplot is found near the level of ‘0.5′ for the y-axis. In our case, all boxplots are noted near the 0.5 mark. In the present analysis, a cumulative gains chart was also presented wherein ‘target’ (total number of cases) and ‘gained’ (in reference to the target of the overall figure of cases for a particular class) were shown. In addition to points along both axes, a ‘baseline’ curve is indicated in the shape of the diagonal line and curve laid above the baseline and is accepted as a greater gain. The lift chart provides a different view of the cumulative gains chart. The values along the y-axis represent the ratio of the cumulative gains for each curve (category/ subgroups) against the baseline curve. In this way, the lift at 10% for category AML is 25%/10% = 2.5. Both cumulative and lift charts are based on combined testing and training samples.

## 4. Discussion

The Research CBC parameters (CPD items) generated by modern hematology analyzers are promising automated quantitative morphological parameters (by quantifying the changes in morphological characteristics of blood cells) to introduce as signatory CBC-based diagnostic items for hematological neoplasms, especially leukemias. The potential clinical utility of CPD items for early exclusion of hematological disorders is well reported in various studies [21,22,25]. However, none of these studies applies AI tools and only a few studies report the utility of these CPD parameters by aiding statistical equations and simple models/classifiers. As an illustration, a study by Yang et al. evaluated CPD (white cell scattering) items for differentiation of acute leukemia lineage by using the Coulter DxH800 analyzer. In this study, the authors derive a 21-CBC items-based model (generated by the KYL program, an Excel macro program) and reported very high (100%) specificity and sensitivity for differentiation of acute promyelocytic leukemia (APL) cases, while for ALL, comparably less significant specificity and sensitivity was achieved [33]. In another study by Virk et al., the clinical utility of the CPD parameters, scatter grams, and flags by generating statistical equations for screening of AML cases was published [34]. Here, an exception is found in studies conducted by Shabbir et al. [35] and our work (Haider et al.) [36], wherein ML tools were challenged for discernment between just two study groups (hematologic vs. non-hematologic malignancies and acute promyelocytic leukemia vs. other hematologic malignancies, respectively).

The findings from our study’s dataset present a promise for this early pre-microscopic prediction of leukemias and related common hematological neoplasms, since the Research CBC parameters are not only able to detect the presence of leukemia but can assist in prediction of the lineage and type of leukemias. In addition, we showed that an AI approach, using an ML algorithm trained on the Research CBC parameters along with routine items-based results (dataset), is not only able to differentiate the lineage of leukemia (either myeloid or lymphoid) but can also remain predictive for the type (acute or chronic or other) of leukemia and related disorders. The result showed that our RFBN model performed the classification with high accuracy and successfully differentiated the study groups with a significant (10.6 percent) in-correction rate. The utility of our suggested Research CBC parameters-driven predictive model for leukemias and related disorders was established through its practical results of 83.1% and 89.4% for training and testing datasets, respectively (Figure 1). The higher accuracy from RFBN in our case can be justified as RFBN increases the feature vector (hidden layers). When the dimension of hidden layers is increased, the linear reparability of hidden layers increases. The importance of the proposed CBC data-driven ML predictive modeling increases for clinics and/or diagnostic setups not specializing in hematology by serving as a screening tool to aid them in diagnostic procedures and the proper and early referral of patients with hematological emergencies such as acute leukemias. Furthermore, our proposed model can minimize irrational ordering of laboratory tests (usually used for additional confirmations) in order to research the correct diagnosis. AI applications in laboratory diagnostics facilitates early (decisions) diagnosis through a limited profile of laboratory tests [10]. The suggested approach has the potential to be replicated to differentiate other hematological disorders.

The results of this study are appealing and serve to legitimize the in-practice clinical approach. Clinicians have acknowledged that results of laboratory investigations, examined in isolation, have definite diagnostic importance in clinical decision-making, typically in environments where clinical knowledge and expertise play a critical role [11]. This belief is logical considering the deficiency of the reported work on the Research CBC parameters such as CPD items and AI tools in hematology (laboratory) diagnostics. Accordingly, the use of the AI approach in the clinical interpretation of laboratory tests can be more diagnostically valuable as a speedy, directed, and more persuasive practice in lieu of trailing the probable diagnosis through an increasing number of irrational and expensive tests. The utility of AI/ML predictive models can be effectively enhanced provided that information regarding clinical (patient’s) history and physical examination of patients become an integral part by adding more valuable attributes to such models. Here, it is necessary to mention that the scope of our study is limited in its boarder clinical utility. This limitation can be addressed by validation/re-validation, ideally through external independent datasets at multiple centers.

## 5. Conclusions

In the present study, a particular disease’s signature raised by the research CBC parameters is illustrated for early differentiation among various types of leukemias and related disorders. The CPD-driven ANN modeling can be a novel practice that substantially strengthens the disease signatory characteristics from these research CBC items in different types of leukemias. The suggested approach can reduce the frequency of extra and irrational diagnostic test ordering that is not only time-consuming but is also an extra burden on patients and laboratory staff. Here, the application of CPD-driven predictive modeling as an assistant predictive tool in the decision support system of hematology laboratory/clinics is suggested.

## Figures and Tables

**Figure 1 diagnostics-12-00138-f001:**
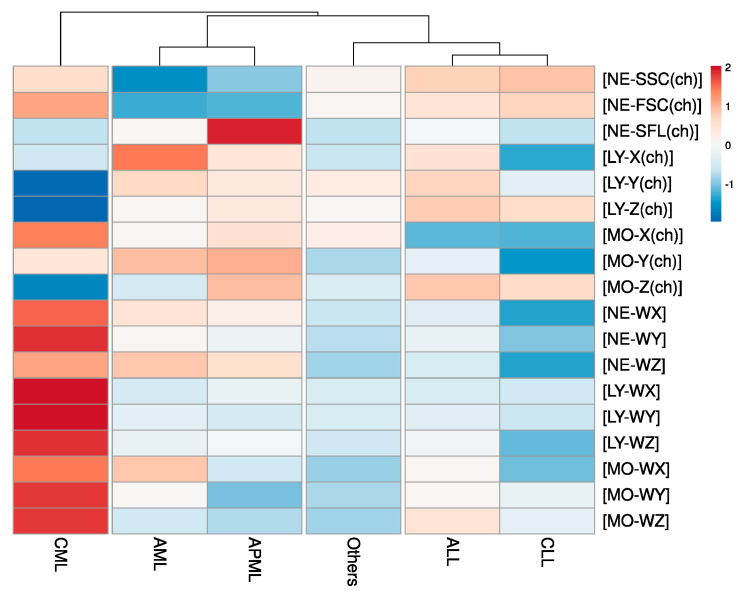
The heat map: color grading and clustering trends of CBC Research parameters among study groups. For heat map color grading ‘diverging Red to Blue’ scheme (for higher to lower values, respectively) was used. The clustering of study groups (columns) is presented on function ‘correlation’.

**Figure 2 diagnostics-12-00138-f002:**
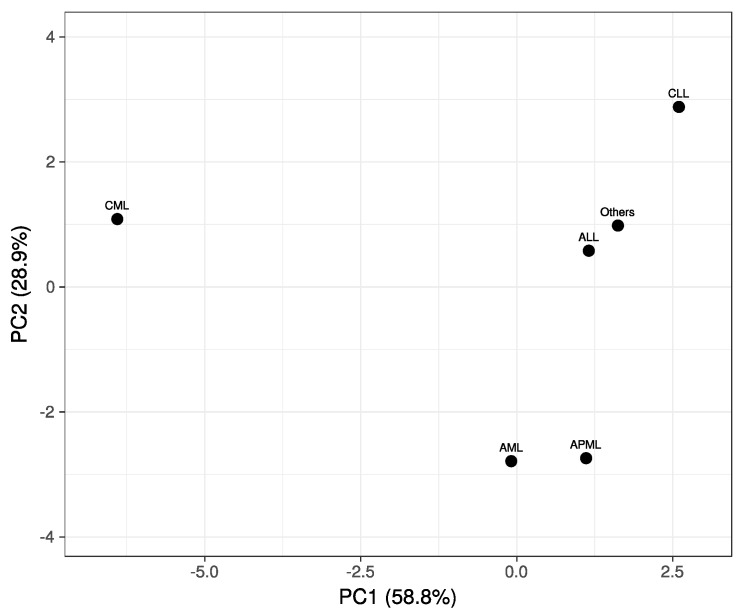
Principal Component Analysis (PCA) plot demonstrating Research CBC parameters driven relatedness among various types of leukemias (our study groups).

**Figure 3 diagnostics-12-00138-f003:**
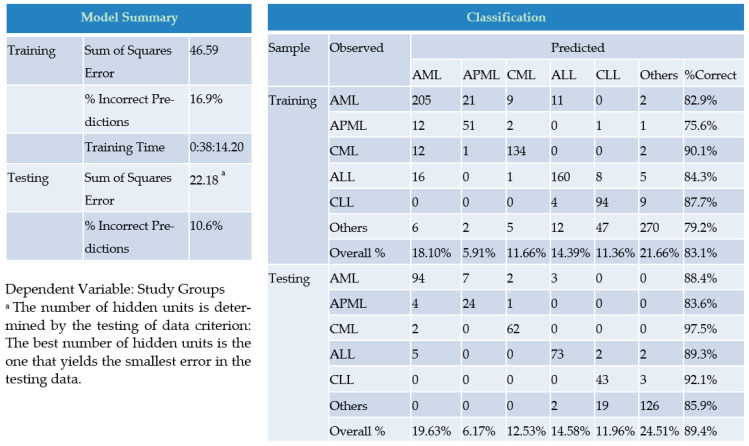
The model summary, classification table, predicted-by-observed chart, ROC curve, cumulative gains and lift chart for the Research CBC parameters driven Radial Basis Function (RBF) predictive model.

**Table 1 diagnostics-12-00138-t001:** Baseline characteristics (classical CBC and research CBC (CPD) of analytic cohorts, according to types of leukemias and related hematological disorders).

Study Parameters	Study Groups						Sig.
	AML	APML	CML	ALL	CLL	Others	
	Mean ± SD	Mean ± SD	Mean ± SD	Mean ± SD	Mean ± SD	Mean ± SD	
Automated Classical CBC Parameters							
Hb	8.19 ± 2.10	8.56 ± 1.61	9.38 ± 1.87	8.17 ± 2.59	10.64 ± 2.47	9.69 ± 3.24	<0.005
RBC (10^12^/L)	2.78 ± 0.82	2.93 ± 0.61	3.49 ± 0.8	3.02 ± 1.29	3.92 ± 0.99	3.68 ± 1.56	<0.005
PCV	25.09 ± 8.09	25.6 ± 5.29	28.86 ± 6.24	24.73 ± 7.64	34.39 ± 7.51	30.27 ± 10.53	<0.005
MCV	90.34 ± 10.47	88.18 ± 8.06	83.45 ± 10.14	83.94 ± 9.12	88.93 ± 9.01	85.47 ± 10.93	<0.005
MCH	29.39 ± 3.53	29.01 ± 3.05	27.1 ± 3.9	27.36 ± 2.91	27.41 ± 3.29	27.15 ± 4.18	<0.005
MCHC	32.35 ± 1.93	32.95 ± 2.34	32.15 ± 2.25	32.64 ± 1.93	30.81 ± 2.44	31.65 ± 1.92	<0.005
WBC (10^9^/L)	39.66 ± 66.75	26.8 ± 47.65	192.39 ± 142.46	70.91 ± 107.47	95.81 ± 123.45	16.99 ± 36.84	<0.005
PLT (10^3^/μL)	60.88 ± 83.18	53.73 ± 85.94	438.42 ± 292.94	53.74 ± 62.92	187.03 ± 105.8	304.35 ± 406.31	<0.005
NEUT# (10^3^/μL)	9.59 ± 29.59	10.35 ± 18.81	161.65 ± 125.42	3.25 ± 4.22	5.83 ± 4.49	8.39 ± 13.65	<0.005
LYMPH# (10^3^/μL)	9.07 ± 12.93	4.77 ± 11.01	9.62 ± 5.23	47.76 ± 77.4	82.74 ± 115.79	6.19 ± 31.27	<0.005
MONO# (10^3^/μL)	21.29 ± 42.93	12.05 ± 25.11	8.25 ± 8.81	20.09 ± 42.26	6.8 ± 17	1.91 ± 5.26	<0.005
EO# (10^3^/μL)	0.18 ± 0.99	0.07 ± 0.15	5.1 ± 5.24	0.13 ± 0.29	0.3 ± 0.42	0.36 ± 1.35	<0.005
BASO# (10^3^/μL)	0.07 ± 0.21	0.06 ± 0.13	5.32 ± 5.22	0.15 ± 0.39	0.15 ± 0.23	0.08 ± 0.15	<0.005
NEUT (%)	22.16 ± 19.91	35.91 ± 19.93	81.92 ± 11.61	13.12 ± 17.24	12 ± 12.26	54.22 ± 23.9	<0.005
LYMPH (%)	37.9 ± 22.28	37.91 ± 26.65	7.19 ± 5.86	64.36 ± 22.07	80.56 ± 17.37	31.97 ± 22.22	<0.005
MONO (%)	39.09 ± 23.61	25.37 ± 21.72	4.88 ± 4.61	21.07 ± 18.04	6.48 ± 10.98	11.35 ± 11.68	<0.005
EO (%)	0.67 ± 2.07	0.71 ± 1.5	3.16 ± 5.5	0.5 ± 0.96	0.73 ± 1.5	1.96 ± 2.45	<0.005
BASO (%)	0.18 ± 0.36	0.1 ± 0.16	2.85 ± 2.04	0.22 ± 0.3	0.23 ± 0.33	0.49 ± 0.75	<0.005
IG# (10^3^/μL)	1.86 ± 4.83	1.53 ± 3.76	65.04 ± 57.27	0.73 ± 1.51	0.45 ± 1.62	1.18 ± 3.61	<0.005
IG (%)	4.38 ± 6.33	5.08 ± 7.77	30.31 ± 9.61	1.76 ± 2.93	0.54 ± 1.33	4.07 ± 6.64	<0.005
NRBC# (10^3^/μL)	0.35 ± 1.07	0.11 ± 0.22	2.16 ± 3.42	0.51 ± 1.56	0.05 ± 0.27	0.47 ± 4.46	<0.005
NRBC (%)	1.61 ± 3.55	0.91 ± 1.35	1.15 ± 1.38	1.4 ± 3.87	0.28 ± 1.51	1.56 ± 8.43	0.549
PDW (fL)	8.76 ± 7.24	6.19 ± 7.35	11.09 ± 6.35	7.03 ± 6.72	11.92 ± 4.6	8.31 ± 6.58	<0.005
MPV (fL)	7.23 ± 5.46	5.05 ± 5.67	8.83 ± 4.6	5.89 ± 5.35	9.93 ± 3.39	6.97 ± 5.25	<0.005
PCT (%)	0.05 ± 0.09	0.04 ± 0.09	0.41 ± 0.35	0.04 ± 0.07	0.19 ± 0.12	0.28 ± 0.42	<0.005
Retic count	1.93 ± 11.84	1.15 ± 1.51	3.13 ± 2.21	0.55 ± 1.07	0.2 ± 0.54	1.88 ± 1.63	0.015
**Automated Research CBC (CPD) Parameters**							
NE–SSC(ch)	140.81 ± 14.05	143.08 ± 10.87	149.05 ± 6.53	149.64 ± 9.26	150.16 ± 7.89	147.15 ± 10.04	<0.005
NE–SFL(ch)	51.43 ± 17.33	65.85 ± 22.71	45.86 ± 5.12	50.71 ± 8.3	45.81 ± 8.45	45.66 ± 7.27	<0.005
NE–FSC(ch)	72.29 ± 11.15	72.59 ± 11.81	84.04 ± 5.57	80.89 ± 7.03	82.23 ± 5.73	78.92 ± 7.69	<0.005
LY–X(ch)	87.33 ± 10.39	84.5 ± 10.35	81.63 ± 8.89	84.75 ± 7.25	79.58 ± 4.46	81.45 ± 4.49	<0.005
LY–Y(ch)	68.65 ± 12.29	65.54 ± 9.37	42.89 ± 19.68	68.91 ± 16.15	59.04 ± 8.9	65.11 ± 6.06	<0.005
LY–Z(ch)	56.68 ± 3.74	57.32 ± 3.02	52.44 ± 3.49	58.2 ± 3.79	57.78 ± 2.94	56.66 ± 2.39	<0.005
MO–X(ch)	118.05 ± 8.27	120.75 ± 9.83	126.3 ± 6.91	110.2 ± 7.39	109.97 ± 6.14	119.14 ± 5.74	<0.005
MO–Y(ch)	114.65 ± 23.51	115.35 ± 25.35	112.09 ± 24.26	108.43 ± 23.79	101.6 ± 9.56	105.47 ± 17.45	<0.005
MO–Z(ch)	62.66 ± 4.97	65.49 ± 7.92	60.28 ± 2.89	65.29 ± 6.54	64.9 ± 3.53	62.82 ± 4.76	<0.005
NE–WX	435.71 ± 127.01	419.16 ± 119.61	501.29 ± 76.69	386.73 ± 108.58	323.69 ± 61.47	368.47 ± 88.09	<0.005
NE–WY	1388.88 ± 755.01	1262.53 ± 829.7	2467.69 ± 693.2	1226.47 ± 616.41	740.42 ± 279.96	897.1 ± 471.45	<0.005
NE–WZ	825.5 ± 257.67	801.79 ± 213.15	847.02 ± 109.49	721.08 ± 203.64	650.14 ± 154.81	691.02 ± 156.02	<0.005
LY–WX	533.66 ± 118.75	550.86 ± 136.81	695.52 ± 168.56	535.53 ± 119.29	530.33 ± 115.78	536.78 ± 109.45	<0.005
LY–WY	1069.66 ± 267.76	994.91 ± 184.93	1929.71 ± 1070.73	1060.03 ± 231.82	960.37 ± 169.92	1007.77 ± 220.04	<0.005
LY–WZ	568.06 ± 115.83	586.67 ± 142.48	801.74 ± 165.36	578.5 ± 138.35	460.95 ± 102.18	527.32 ± 122.95	<0.005
MO–WX	340.51 ± 75.02	301.81 ± 104.41	357.22 ± 65.23	319.04 ± 90.03	285.66 ± 66.46	291.38 ± 73.36	<0.005
MO–WY	873.84 ± 282.05	701.67 ± 446.57	1146.88 ± 346.87	878.07 ± 317.66	832.36 ± 218.58	736.74 ± 258.89	<0.005
MO–WZ	616.05 ± 112.94	601.16 ± 204.8	767.94 ± 100.79	681.88 ± 226.76	636 ± 255.98	597.25 ± 156.62	<0.005

Hb; hemoglobin, RBC; red blood cell, PCV; packed cell volume, MCH; mean cell hemoglobin, MCHC; mean cell hemoglobin, WBC; white blood cell, PLT; platelet, NEUT# (10^3^/μL); absolute neutrophil, LYMPH# (10^3^/μL); absolute lymphocyte count, MONO# (10^3^/μL); absolute monocyte count, EO# (10^3^/μL); absolute eosinophil count, BASO# (10^3^/μL); absolute basophil count, NEUT (%); percent neutrophil count, LYMP (%); percent lymphocyte count, MONO (%); percent monocyte count, EO (%); percent eosinophil count, BASO (%); percent basophil count, IG# (10^3^/μL); absolute immature granulocyte count, IG (%); percent immature granulocyte count. NE-SSC; neutrophil side scatter, NE-SFL; neutrophil side fluorescence, NE-FSC; neutrophil forward scatter, LY-X; lymphocyte side scatter, LY-Y; lymphocyte side fluorescence, LY-Z; lymphocyte forward scatter, MO-X; monocyte side scatter, MO-Y; monocyte side fluorescence, MO-Z; monocyte forward scatter, NE-WX; neutrophil side scatter distribution width, NE-WY; neutrophil side fluorescence distribution width, NE-WZ; neutrophil forward scatter distribution width, LY-WX; lymphocyte side scatter distribution width, LY-WY; lymphocyte side fluorescence distribution width, LY-WZ; neutrophil forward scatter distribution width, MO-WX; monocyte side scatter distribution width, MO-WY; monocyte side fluorescence distribution width, MO-WZ; monocyte forward scatter distribution width.

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
