# Peer review of "Beyond the In-Practice CBC: The Research CBC Parameters-Driven Machine Learning Predictive Modeling for Early Differentiation among Leukemias"

_diagnostics, 2022, doi:10.3390/diagnostics12010138_

Round 1

Reviewer 1 Report

Although the revised paper has many limitations, I agree that it is acceptable.

Nothing to add

Author Response

Great thanks for your kind reviewer and suggestions in first round.

We valued and tried our best to addressed all suggestions/comments from both reviewers.

Reviewer 2 Report

The manuscript has been revised for better with additional data and accompanied statistics. I have a minor suggestion to improve upon the PCA Plot.

The authors shall consider the 'Biplot' (scores + loadings), rather than plotting only the scores. Then PCA plot ( fig 2) can then be easily compared with the heat map (fig 1).

Additionally, if the raw data had been subjected to PCA  using feature value as it is (i.e without averaging for each subset of leukemia cases), then we shall see how well the 'scores' are scattered among the clusters of different groups.

The authors shall consider providing the software (as an open source) for other medical professionals to try and use for diagnostic purposes. (A download weblink can be provided)

Author Response

Great thanks for these meaningful suggestions.

We respect and valued the shared comments. Regarding PCA plot, we created this through a Clustvis; a web tool for visualization of multivariate data (https://biit.cs.ut.ee/clustvis/). At the platform of this web tool, it enough convenient for other scientist to replicate our suggested exercise and they can get valuable visualization, too. But unfortunately, the suggested web tool has allowed some limited option for presentation of dataset into PCA, 'Biplot' (scores plus loading) is one of them.

As per your kind comment, in method section -data analysis part the aforementioned details and link for this web tool 'Clustvis' is added in this revised version.   

This manuscript is a resubmission of an earlier submission. The following is a list of the peer review reports and author responses from that submission.

Round 1

Reviewer 1 Report

This is a paper on the challenging development of leukemia diagnosis using AI. Although I am not an AI expert, it is said that large, proper, and correct data should be input for machine learning for the development of diagnostic AI. The laboratory data of acute leukemia vary greatly depending on the subtype and patients’ age. Many benign and malignant diseases should be differentiated. Unfortunately, the subject of this study is only healthy control, AML, and ALL, which background remains unknown.

It is also necessary to perform verification using another cohort (preferably at another medical institution).

Line 17 & others, CBC is usually an abbreviation for “complete blood count”.

Line 130, How many cases for learning, and how many for study?

Table 1, RBC in ALL:5.29 million?

Table 1 & Table 2, the parameter [NE-SSE] and others should be explained, because the readers are not specialist of XN-1000.

Line 297, XX?, XY?

Author Response

Answers/Response

Thank you for raising key comments. The extensive modification in accordance to suggestions from reviewers is completed in this revised version. The subject of present study was to propose the CBC/CPD data driven AI tools for differentiation of acute leukemia lineage (myeloid or lymphoid) in hematology-oncology labs/clinics where expert morphologist, cytochemical stains and related tests like Flowcytometery (phenotyping) are not under flag of routine hematology laboratory. The suggested approach is ‘pre-microscopic’ and ‘handy’; as in every routine laboratory these details (parameters) are routinely generated without any extra effort. The healthy control group was added aimed sharing a normal range (obtained in our Pakistani community) and just contrast against disease groups.

Under limitation of present study and recommendation paragraph, need of external independent dataset validation (verification) is suggested in this revised version.

CBC abbreviation is corrected.

Cases for learning and training (study) were distributed in three schemes 50, 60, 70 and 50, 40, 30 to get better results. Otherwise most of studies focused on 70:30 scheme.

RBC in ALL is 3.29.

Thanks for valuable comment. Abbreviations used in table 1 and 2 are expanded in footnote.

Reviewer 2 Report

The article titled " Yonder the in-practice CBC: Machine Learning based decipherment of Research CBC Parameters yoked with Acute Leukemia and underlying effected Lineage’ discernment by Rana et Al was reviewed and the suggestions are as follows: 

1) Rephrase 'clinical personals' to 'clinicians' or 'clinical staffs' etc..

2) Pg2, ln 81: The following sentence is not simple enough and require paraphrasing:
The hypothesis of present study was that the “signature” of common hematological neoplasms particularly acute leukemias raised in the values of CBC data (conventional and CPD items) enough  aspiring for ML based predictive modeling in differentiate of probable diagnosis

3) How the authors came upon 5% for disagreement? shall add a line explaining it: 
"The opinion of a third hematologist was only asked when results had >5% disagreement."

4) ‘Homogenous subsets’: briefly explain or give reference, what do you mean by this.

5) pg3, ln132: better or superior? how much in terms of a quantified measure?

6) All abbreviations used in Table one should be expanded atleast once either in the manuscript or put together in supporting info (SI). It is better to define and brief these abbreviations for conveying its significance in ALL, AML diagnosis.
7) Syntax issues:
pg.4:'NEUT% (%)' -> NEUT (%)
pg.6:'% Promyelocytes' -> Promyelocytes(%)
pg.6: Do you mean 'plus' here?: 'subsets as AML vs. ALL pulse control'

7) Why REBN performs superior to other ML tools should be discussed in the discussion section.

8) Mention and discuss on the  similar works on Leukemia using ML carried out earlier.

Ref: CPD based AI: Artificial Intelligence based Models for Screening of Hematologic Malignancies using Cell Population Data; Shabbir Syed-Abdul, Rianda-Putra Firdani, Hee-Jung Chung, Mohy Uddin, Mina Hur, Jae Hyeon Park, Hyung Woo Kim, Anton Gradišek & Erik Dovgan 

Ref: Cell Population Data–Driven Acute Promyelocytic Leukemia Flagging Through Artificial Neural Network Predictive Modeling Rana Zeeshan Haider,∗‡∗ Ikram Uddin Ujjan,† and Tahir S. Shamsi∗

Ref: Machine Learning in Detection and Classification of Leukemia Using Smear Blood Images: A Systematic Review Mustafa Ghaderzadeh ,1 Farkhondeh Asadi ,1 Azamossadat Hosseini ,1 Davood Bashash ,2 Hassan Abolghasemi ,3 and Arash Roshanpour4

9) Is the algorithm capable of differentiating  chronic luekemia (and its lineages) cases too .. ? It is good to comment on it in the discussion part.

Author Response

Comments to the Author

The article titled “Yonder the in-practice CBC: Machine Learning based decipherment of Research CBC Parameters yoked with Acute Leukemia and underlying effected Lineage’ discernment” by Rana et al was reviewed and the suggestions are as follows;

Suggestion-1) Rephrase ‘clinical personals’ to ‘clinicians’ or ‘clinical staffs’ etc.

Answer/Response: Thanks for your valuable comments. Yes, the suggested word replacement is done in this revised version.

Suggestion-2) Pg2. In 81: The following sentence is not simple enough……

Answer: For better understanding, paraphrasing of mentioned sentence is done.

Suggestion-3) How the authors came upon 5% for disagreement?......

Answer: For explanation a line is added.

Suggestion-4) ‘Homogenous subsets’: briefly explain…..

Answer: Definition and technical details are added.

Suggestion-5) pg3. In132 better or superior? ….

Answer: Along with word replacement and brief explanation is provided.

Suggestion-6) All abbreviation used in Table one should be expanded….

Answer: Thanks for kind comment. Yes, In footnote of table-1 all abbreviations are expanded.

Suggestion-7) Syntax issues: pg.4: NEUT%(%)……

Answer: Syntax issues are resolved in this corrected version.

Suggestion-7) Why RFBN performed superior to other ML Tools?....

Answer: The working/technical advantages of RFBN over other ML Tools are discussed in discussion section.

Suggestion-8) Mention and discuss on the similar works on leukemia using ML carried out earlier.

Answer: Nice comment and thanks for providing us examples. Yes, the studies based CPD/CBC data driven ML Tools methodology are discussed in introduction and discussion section.

Suggestion-9) Is the algorithm capable of differentiation chronic leukemia (and its lineages) cases to ….?

Answer: Great suggestion. Yes, we briefly added and suggested this point in discussion section.